# Influence of Different Pt Functionalization Modes on the Properties of CuO Gas-Sensing Materials

**DOI:** 10.3390/s24010120

**Published:** 2023-12-25

**Authors:** Xiangxiang Chen, Tianhao Liu, Yunfei Ouyang, Shiyi Huang, Zhaoyang Zhang, Fangzheng Liu, Lu Qiu, Chicheng Wang, Xincheng Lin, Junyan Chen, Yanbai Shen

**Affiliations:** 1Zijin School of Geology and Mining, Fuzhou University, Fuzhou 350108, China; cxxneu@outlook.com (X.C.); thliufzu@163.com (T.L.); 15170743807@163.com (Y.O.); 13950791996@163.com (S.H.); zhangzhaoyang127@126.com (Z.Z.); 13705069891@163.com (F.L.); 15805909779@163.com (L.Q.); 17359862692@163.com (C.W.); 18006936326@163.com (X.L.); 17759731504@163.com (J.C.); 2Fujian Key Laboratory of Green Extraction and High Value Utilization of New Energy Metals, Fuzhou 350108, China; 3School of Resources and Civil Engineering, Northeastern University, Shenyang 110819, China

**Keywords:** gas sensor, CuO, Pt doping, Pt loading, NO_2_

## Abstract

The functionalization of noble metals is an effective approach to lowering the sensing temperature and improving the sensitivity of metal oxide semiconductor (MOS)-based gas sensors. However, there is a dearth of comparative analyses regarding the differences in sensitization mechanisms between the two functionalization modes of noble metal loading and doping. In this investigation, we synthesized Pt-doped CuO gas-sensing materials using a one-pot hydrothermal method. And for Pt-loaded CuO, Pt was deposited on the synthesized pristine CuO surface by using a dipping method. We found that both functionalization methods can considerably enhance the response and selectivity of CuO toward NO_2_ at low temperatures. However, we observed that CuO with Pt loading had superior sensing performance at 25 °C, while CuO with Pt doping showed more substantial response changes with an increase in the operating temperature. This is mainly due to the different dominant roles of electron sensitization and chemical sensitization resulting from the different forms of Pt present in different functionalization modes. For Pt doping, electron sensitization is stronger, and for Pt loading, chemical sensitization is stronger. The results of this study present innovative ideas for understanding the optimization of noble metal functionalization for the gas-sensing performance of metal oxide semiconductors.

## 1. Introduction

Nitrogen dioxide (NO_2_) is a prevalent air pollutant that can cause significant environmental issues, such as acid rain and the depletion of the ozone layer. It can also harm the respiratory system, eyes, skin, and other organs of the human body, and severe cases can lead to respiratory disorders or death. To protect humans and the environment from the ill effects of continuous exposure to NO_2_ and to detect whether emission concentrations exceed permissible limits, the real-time monitoring of NO_2_ is vital [1,2,3].

Due to their high sensitivity, low cost, and fast response, semiconductor sensors have been widely applied for monitoring toxic and hazardous gases. Metal oxide semiconductor (MOS) materials, which have abundant active adsorption sites [4] and demonstrate sensitive resistance changes in response to changes in the environmental gas atmosphere [5], have been extensively studied and utilized as gas-sensing materials. In particular, n-type metal oxide semiconductors have exhibited excellent gas-sensing performance for NO_2_. For example, Lv et al. [6] used willow catkins as a biomass template to obtain thin ZnO nanorods via immersion and calcination treatment, achieving a response of 100.4 for 10 ppm NO_2_ at 92 °C. Similarly, Gajanan et al. [7] prepared WO_3_ nanosheets via a hydrothermal method, achieving a response of 2.25 for 100 ppm NO_2_ at 200 °C. Patil et al. [8] synthesized In_2_O_3_ nanoparticle assemblies as nano-cubes via a simple hydrothermal method, achieving a response of 10 for 3 ppm NO_2_ at 50 °C with a response time of 21 s, a recovery time of 8.7 min, and a detection limit of 60 ppb. In recent years, p-type CuO gas-sensitive materials have been increasingly utilized in gas sensors due to their excellent chemical stability and catalytic activity [9] and have been widely applied for detecting reducing gases such as ethanol [10], hydrogen [11], and carbon monoxide [12]. However, research on oxidizing gases such as NO_2_ remains scarce. Y. H. Navale et al. [13] synthesized a CuO nanofilm through thermal evaporation, achieving a response of 76% for 100 ppm NO_2_ at 150 °C, demonstrating good selectivity toward NO_2_ but with poor recovery. Han et al. [14] prepared CuO nanowires (NWs) via thermal oxidation, achieving a response of 1.58 for 100 ppm NO_2_ at 250 °C with response and recovery times of 60/225 s. Clearly, pristine CuO exhibits inferior performance compared to the n-type MOS materials mentioned above in detecting NO_2_ gas.

To enhance the gas-sensing performance of pristine CuO semiconductor materials, researchers have explored multiple methods, including designing new nanostructure morphologies [15], constructing heterojunctions [14], and functionalizing the materials with noble metals [16]. These approaches aim to address the issues of high operating temperatures and low sensitivity. Among these methods, functionalization with noble metals has been proven to be effective in improving the gas-sensing performance of MOS materials in low-temperature and high-humidity environments. Two mechanisms, electronic sensitization and chemical sensitization, were proposed by Yamazoe et al. [17,18] and Morrison [19] to explain the sensitization effect of noble metal functionalization on the gas-sensing performance of MOS materials. Currently, these mechanisms of noble metal sensitization are widely recognized and applied in practical gas-sensing applications.

Noble metal functionalization of MOSs can be categorized into two types, loading and doping, depending on the timing of functionalization. Loading refers to the functionalization of the prepared MOS with noble metals, causing small noble metal phases to load onto the surfaces of MOS crystals [20]. Regarding noble metal loading on CuO, Chen et al. [21] prepared Au-loaded CuO nanorods by immersing CuO in chloroauric acid. The sensor showed good sensitivity and selectivity toward NO_2_ at room temperature, with a response of 4.3 to 5 ppm NO_2_ at 100 °C. Similarly, Jun-Seong Lee et al. [22] deposited a gold layer on the surface of CuO via sputtering and heat treatment, producing Au-nanoparticle-loaded CuO nanowires. This resulted in the significant enhancement of the response of CuO to NO_2_ at 300 °C due to the presence of 60 nm sized Au nanoparticles. Jae Eun Lee et al. [23] successfully modified the surface of CuO with Pt nanoparticles using chloroplatinic acid and ultrasonic treatment, resulting in the enhanced selectivity and sensitivity of CuO toward HCHO at 225 °C, with a response of 4.31 to 1 ppm HCHO. In contrast, doping refers to the implantation of metal ions into the lattice of a semiconductor [24,25], with noble metal particles appearing inside and/or on the surfaces of MOS crystals during their formation [20]. There have been limited studies on the noble metal doping of CuO, with Jyoti et al. [26] using a hydrothermal method to prepare Ag-CuO nanobricks by adding silver nitrate to the precursor solution. The addition of Ag resulted in more oxygen vacancies and the increased sensitivity of Ag-CuO to NO_2_ at room temperature, with a response of 67.2% for 20 ppm NO_2_ and response and recovery times of 35 s and 900 s, respectively. While the enhancement of CuO gas sensitivity by both noble metal functionalization modes can be explained by the sensitization mechanism mentioned above, the induced lattice changes result in some differences in the promotion mechanism of the gas sensitivity of CuO between the different functionalization modes. However, there is currently a lack of research in this area.

The present work focuses on the synthesis of CuO gas-sensing materials with two noble metal functionalization modes, Pt loading and doping, using a hydrothermal method. The effects of these two functionalization modes on the enhancement of CuO sensitivity to NO_2_ were compared using identical morphological structures. The materials were characterized using XRD, SEM, TEM, and XPS. The gas sensitivity test results indicate that both Pt loading and doping can effectively improve the sensitivity and selectivity of CuO to NO_2_ at low temperatures, with a good room-temperature response. However, the sensing performance of Pt-loaded CuO was better at room temperature, while Pt-doped CuO exhibited lower room-temperature resistance and more significant response changes as the temperature rose. The sensing mechanisms of these two functionalization types of CuO gas-sensing materials to NO_2_ were explored based on the theory of chemical and electronic sensitization.

## 2. Materials and Methods

### 2.1. Material Preparation

The chemicals used in this work, including sodium hydroxide (NaOH), copper sulfate pentahydrate (CuSO_4_·5H_2_O), trisodium citrate dihydrate (C_6_H_5_O_7_Na_3_·2H_2_O), and absolute ethanol (C_2_H_5_OH), were purchased from Sinopharm Chemical Reagent Co., Ltd. Chloroplatinic acid (H_2_PtCl_6_·6H_2_O) was purchased from Chengdu Xiya Chemical Co., Ltd. All of the chemicals used in the experiment were used without further purification. Deionized water was used throughout the whole experimental process.

Pristine CuO nanomaterials and Pt-doped CuO nanomaterials were prepared by using a hydrothermal method. First, a 1 mol/L NaOH solution was added dropwise to a H_2_PtCl_6_·6H_2_O solution until the pH of the solution reached 7.0. Then, 1.3 mmol of CuSO_4_·5H_2_O and 0.91 mmol of C_6_H_5_O_7_Na_3_·2H_2_O were dissolved in 40 mL of deionized water, and after stirring for 15 min at room temperature, 5.3 mmol of NaOH was added to the solution and stirred for 0.5 h. A certain amount of the above H_2_PtCl_6_·6H_2_O solution was added to the solution and stirred for another 2 h. The mixture was then transferred to a 200 mL PTFE-lined autoclave and kept at 160 °C for 12 h. After the reaction was completed and cooled to room temperature, the product was collected by centrifugation, washed with deionized water and absolute ethanol, and dried at 60 °C. The product was then placed in a tube furnace and annealed at 400 °C for 4 h at a heating efficiency of 10 °C/min to obtain 0 mol% (pristine CuO), 1 mol%, 2 mol%, and 5 mol% Pt-doped CuO. The Pt doping concentrations were obtained by calculating the molar ratios of the added H_2_PtCl_6_·6H_2_O to CuSO_4_·5H_2_O.

Pt-loaded CuO nanomaterials were obtained by using an impregnation method. A certain amount of the H_2_PtCl_6_·6H_2_O solution was mixed with 50 mL of absolute ethanol, and 1 mol/L NaOH solution was added dropwise to the solution until the pH of the solution reached 9.0. Then, 0.02 g of the unannealed pristine CuO nanomaterial was added to the solution and heated and stirred at 80 °C until the solution evaporated. The obtained dry product was washed with deionized water and absolute ethanol. The rest of the experimental procedure was the same as that for the preparation of pristine CuO nanomaterials and Pt-doped CuO nanomaterials. The Pt-loading concentrations were obtained by calculating the molar ratio of the added H_2_PtCl_6_·6H_2_O to the unannealed pristine CuO nanomaterials and were 1 mol%, 2 mol%, and 5 mol%, respectively.

For ease of description, the 1 mol% Pt, 2 mol% Pt, and 5 mol% Pt-doped CuO nanorods are abbreviated as 1 mol% Pt-CuO, 2 mol% Pt-CuO, and 5 mol% Pt-CuO. The 1 mol% Pt, 2 mol% Pt and 5 mol% Pt-loaded CuO nanorods are abbreviated as 1 mol% Pt@CuO, 2 mol% Pt@CuO, and 5 mol% Pt@CuO, respectively.

### 2.2. Characterization of the Samples

The compositions and crystal structures of the samples were investigated with an X-ray diffractometer (XRD, PANalytical X’Pert Pro, Cu Kα1 radiation, λ = 1.5406 Å, 2θ range of 10–90°). A field-emission scanning electron microscope (FESEM, ZEISS Ultra Plus, accelerating voltage 20 kV) equipped with an energy-dispersive X-ray spectroscope (EDS) was used to observe the microstructures and sizes of the samples and analyze the element compositions. For further microstructure observation, the size and crystal structure were studied using a TEM (Philips CM200, accelerating voltage o9f 200 kV). Chemical bonding and functional groups were analyzed by using FT-IR (NICOLET 380, wavenumber region of 4000–400 cm^–1^). An X-ray photoelectron spectrometer (XPS, EscaLab 250Xi, energy step size of 0.06 eV) was used to investigate the chemical compositions and valence states of the elements in the samples. All of the binding energies were calibrated with the saturated hydrocarbon C 1s peak at 284.8 eV. A PerkinElmer Lambda 950 UV–vis spectrometer was used to measure the reflectance spectra. The band gaps of the samples can be calculated based on the reflectance spectra.

### 2.3. Fabrication of Gas Sensor and Testing Method

The three samples of pristine CuO, Pt-doped CuO, and Pt-loaded CuO were each mixed with a small amount of deionized water and put into an agate mortar to grind them and form a slurry. Then, the slurry was coated on a tube-sensing electrode, and a nickel-chromium (Ni-Cr) coil through the tube-sensing electrode was used as the heating wire.

The gas-sensing tests were performed on a WS-30A test system (Winsen Electronics Technology Co., Ltd., Zhengzhou, China). The test gases were injected into a closed chamber (18 L) filled with air to obtain the gas-sensitive response. NO_2_ and H_2_ were directly injected by using the static gas distribution method. The vapor of ethanol was prepared by injecting a calculated volume of pure ethanol liquid into the evaporation device in the test chamber and then evaporating it at a temperature over its boiling point. For the preparation of HCHO, SO_2_, and NH_3_ vapors, calculated volumes of their aqueous solutions were also evaporated in the evaporation device in the test chamber. The liquid volume of the target analyte is calculated according to the following formula:$$Q=\frac{C_{g}\times M\times 22.4}{C\times V_{m}}$$
where Q (mL) is the liquid volume of the analyte, $C_{g}$ (ppm) is the required vapor volume fraction, M (g/mol) is the molecular weight of the analyte, C (g/mL) is the concentration of the analyte solution, and $V_{m}$ (mL) is the volume of the testing chamber (18 L). The test operating temperature range was 25–250 °C. Prior to a normal gas-sensing test, the humidity of the chamber was first adjusted to 30% RH, and then the operating temperature of the gas sensor was adjusted to the set value. In the test, the resistance of the sensor can be calculated by measuring the output voltage of the load resistor. The sensor responses to the test gases are defined as R_a_/R_g_. The times required for the sensor to reach a 90% change in resistance after the introduction and depletion of the detection gas are defined as the response and recovery times, respectively.

## 3. Results

### 3.1. Structural and Morphological Characteristics

Figure 1 shows the XRD patterns of pristine CuO, 2 mol% Pt-CuO nanorods, and 2 mol% Pt@CuO nanorods. In Figure 1a, all of the XRD diffraction peaks in the 2 mol% Pt-CuO pattern can be ascribed to the monoclinic crystal system structure of CuO (JCPDS No. 05-0661), and no Pt-related diffraction peaks are found, which is mainly due to the low doping concentration of Pt. In addition to the diffraction peak of the monoclinic crystal system structure of CuO in the plot of 2 mol% Pt@CuO, the (111) crystal plane diffraction peak of the cubic crystal system structure of Pt (JCPDS No. 04-0802) is also present. It can be seen in Figure 1b that in the doping mode, the Pt doped into the lattice of CuO leads to different shifts in the triple high-intensity diffraction peaks in the spectrum of CuO [27,28]. In the loading mode, Pt is not loaded into the CuO lattice, so it will not cause a shift in the CuO diffraction peak [29].

The surface morphology of 2 mol% Pt@CuO nanorods was observed with an SEM, which is shown in Figure 2a,b. After Pt loading, granular products appear on the surfaces of CuO nanorods. According to the energy spectrum of Site A in Figure 2e, it can be inferred that they are Pt nanoparticles. The surface morphology of 2 mol% Pt-CuO nanorods is shown in Figure 2c,d. By comparing them to 2 mol% Pt@CuO nanorods, it can be seen that the morphology of the CuO nanorods is changed after Pt doping, as their length is shortened to 100–150 nm, while their diameter is not affected, but no obvious noble metal particles are observed on their surfaces.

The microscopic morphologies of 2 mol% Pt@CuO nanorods and 2 mol% Pt-CuO nanorods were further investigated comparatively using a TEM. The TEM images of 2 mol% Pt@CuO nanorods are shown in Figure 3a–c, and it can be seen in Figure 3a that the morphology of the CuO nanorods did not change after Pt loading. The granular products are uniformly distributed on the surfaces of the nanorods, and most of the particles are within 10 nm in diameter, while a few particles are agglomerated together to form larger particles. From the high-magnification TEM images shown in Figure 3b,c, it is clear that these particles are Pt metal particles. The grain plane spacings of 0.25 and 0.27 nm can be attributed to the (002) and (110) crystal planes of the monoclinic structure of CuO, respectively, while the grain plane spacings of 0.19 and 0.22 nm can be ascribed to the (200) and (111) crystal planes of the cubic crystal structure of Pt, respectively. The TEM images of 2 mol% Pt-CuO nanorods are shown in Figure 3d,f, and it can be seen from Figure 3d that the morphology of CuO nanorods was changed after Pt doping, forming a state in which short nanorods and nanoparticles coexist, indicating that the growth orientation of CuO crystals becomes irregular after doping with Pt. This is also in general agreement with the XRD analysis results. The 0.17, 0.27, and 0.46 nm crystal plane spacings shown in Figure 3e,f correspond to the (112), (110), and (100) crystal planes of the monoclinic structure of CuO, while the 0.19 and 0.22 nm crystal plane spacings correspond to the (200) and (111) crystal planes of the cubic structure of Pt, respectively.

The XPS spectra of Pt 4f, Cu 2p, and O 1s of CuO after Pt doping and Pt loading are shown in Figure 4. Figure 4a,d show the Pt 4f spectra of Pt-doped and Pt-loaded CuO. There are two pairs of Pt 4f_5/2_ and Pt 4f_7/2_ peaks in Figure 4a; the peaks at 78.0 and 74.5 eV are attributed to Pt^2+^, while the peaks of 75.8 and 72.5 eV indicate the presence of Pt^0^ [30,31]. In Figure 4d, Pt^2+^ is related to the peak at 77.7 eV, while Pt^0^ is associated with the peaks at 75.6 and 72.3 eV [32]. For Pt@CuO in Figure 4a, Pt^0^ nanoparticles are formed on the surface of CuO, but part of Pt is oxidized to PtO. But for Pt-CuO in Figure 4d, the Pt^2+^ indicates that the Pt element was doped into the CuO crystal lattice. Figure 4b,e show the Cu 2p spectra of Pt-doped and Pt-loaded CuO. Two peaks located around 953 eV (Cu 2p_1/2_) and 933 eV (Cu 2p_3/2_) can be attributed to Cu^2+^ [33]. Meanwhile, multiple satellite peaks of Cu^2+^ appear in the direction of high binding energy. Compared with the Pt 4f and Cu 2p spectra of Pt-doped CuO, it can be seen that the Pt 4f_5/2_ and Pt 4f_7/2_ peaks of Pt-loaded CuO shift in the direction of higher binding energy, while Cu 2p_1/2_ and Cu 2p_3/2_ peaks shift in the direction of lower binding energy. This indicates a strong electron transfer from Pt to Cu on the surface of Pt-loaded CuO at room temperature. Two O 1s peaks located around 529 eV and 531 eV in Figure 4c,f correspond to the CuO lattice oxygen and surface hydroxyl oxygen [34], respectively. Based on the calculation result for relative oxygen content, the relative adsorbed oxygen content on the Pt-loaded CuO surface at room temperature is higher than that on Pt-doped CuO. The decoration of Pt indeed increases the specific surface area of the sensing materials, providing more catalytically active sites for gas diffusion and adsorption [35].

### 3.2. Gas-Sensing Properties

Figure 5 shows the effect of two Pt functionalization modes on the gas-sensing properties of CuO to NO_2_ gas. Figure 5a–c show the relationships between gas-sensing properties and the operating temperature for different proportions (1, 2, 5 mol%) of Pt-loaded CuO and pristine CuO in response to 5 ppm NO_2_ gas. Figure 5a shows the relationship between the response of different proportions of Pt-loaded CuO nanorods to NO_2_ gas and the operating temperature. Pt loading effectively improved the response of CuO nanorods to NO_2_ gas at low operating temperatures, and the peak response of 2 mol% Pt@CuO to NO_2_ gas reached 4.4 at a low temperature of 50 °C. Compared with the peak response of pristine CuO nanorods, which was 4.3 at 150 °C, the response of 2 mol% Pt@CuO samples not only improved slightly but also reduced the required operating temperature to 50 °C. Combined with their response and recovery effects in Figure 5b,c, the response/recovery times of 2 mol% Pt@CuO were 50 and 153 s, respectively, when the response reached 50 °C at the peak of 2 mol% Pt@CuO. These times for pristine CuO nanorods were 35 s and 202 s, respectively, under the same conditions. The results show that the response of 2 mol% Pt@CuO is slightly improved at lower operating temperatures, and the recovery speed is also accelerated. Therefore, from the perspective of the gas response and response/recovery times, the optimal operating temperature of Pt-loaded CuO nanorods is 50 °C. The related response and recovery curves of the sensors based on pristine and Pt-loaded CuO nanorods upon exposure to 5 ppm NO_2_ at various operating temperatures are shown in Appendix A.

Figure 5e,f show the relationships between gas-sensing properties and the operating temperature for different proportions (1, 2, 5mol%) of Pt-doped CuO in response to 5 ppm NO_2_. Figure 5d shows the relationship between the response of Pt-doped CuO nanorods with different proportions to NO_2_ gas and the operating temperature. Similar to the performance effects of Pt loading, Pt doping also effectively improved the response of CuO nanorods to NO_2_ gas at low operating temperatures, and 2 mol% Pt doping showed the best sensitization effect at 100 °C, while it reached the peak response at 11.1. Compared with the peak response of 4.3 of pristine CuO nanorods at 150 °C, Pt doping not only improves the response of CuO nanorods but also reduces the required operating temperature. It can be further concluded that the response of Pt-doped CuO nanorods to NO_2_ gas can be significantly enhanced within a 150 °C operating temperature. Figure 5e,f show the response and recovery effects of Pt-doped CuO nanorods and pristine CuO nanorods. After Pt doping, CuO nanorods still maintain a good response time (less than 80 s) and recovery time (less than 300 s). The response time and recovery time of pristine CuO nanorods at 100 °C are 35 s and 188 s, respectively, while the response time and recovery time of 2 mol% Pt-CuO are reduced to 27 s and 168 s, respectively. This indicates that the response of Pt doping can be improved at lower operating temperatures, and the response and recovery speeds can be accelerated. Therefore, from the perspective of the gas response and response/recovery times, the optimal operating temperature of Pt-doped CuO nanorods is 100 °C. The related response and recovery curves of the sensors based on pristine and Pt-doped CuO nanorods upon exposure to 5 ppm NO_2_ at various operating temperatures are shown in Appendix A. All of the resistances of CuO nanorods decrease at the same operating temperature after Pt doping and Pt loading due to the production of oxygen vacancies by Pt functionalization.

Figure 6 further compares the gas-sensing properties of Pt@CuO, Pt-CuO, and pristine CuO in response to different concentrations of NO_2_ at their respective optimal operating temperatures. Figure 6a shows the response–recovery characteristic curves of pristine CuO and Pt@CuO nanorods with different Pt concentrations in response to NO_2_ gas at their respective optimal operating temperatures. Figure 6c shows the response–recovery characteristic curves of pristine CuO and Pt-CuO nanorods in response to NO_2_ gas with different Pt concentrations at their respective optimal operating temperatures. It can be seen that the range of resistance changes in all gas-sensitive materials gradually increases with the increase in the concentration of NO_2_ gas, and the reversibility of the NO_2_ gas reaction is also shown after discharge. Figure 6b shows the relationship between the response of pristine CuO and Pt@CuO nanorods and the concentration of NO_2_ gas. In general, when the concentration of NO_2_ gas is less than 5 ppm, the response of 2 mol% Pt@CuO is better than that of pristine CuO nanorods. Figure 6d shows the relationship between the response of pristine and Pt-CuO nanorods and the concentration of NO_2_ gas. It can be concluded that Pt doping can improve the response of CuO nanorods to NO_2_ gas within the concentration range shown, and 2 mol% Pt-CuO has the best response improvement effect. By comparing the two, it can be found that Pt@CuO can improve the response to low concentrations of NO_2_ gas, while Pt-CuO can improve the response to NO_2_ gas over a wider range of gas concentrations.

The selectivity of the sensor is also an important parameter for gas sensing [36]. A sensor with good selectivity can avoid the cross-interference of the gas and improve the reliability of the response and the response to the target gas [37]. Figure 7 compares the responses of Pt@CuO, Pt-CuO, and pristine CuO to 5 ppm NO_2_, 1000 ppm H_2_, 1000 ppm C_2_H_5_OH, 1000 ppm NH_3_, 1000 ppm HCHO, and 100 ppm SO_2_ gases at their respective optimal operating temperatures. Figure 7a shows that Pt loading was not able to increase the NO_2_ gas selectivity of CuO, and 1 mol% Pt@CuO and 5 mol% Pt@CuO even reduced the response to NO_2_ and improved the response to SO_2_, resulting in the deterioration of the NO_2_ gas selectivity of CuO. As shown in Figure 7b, compared with pristine CuO nanorods, 1 mol% Pt-CuO and 2 mol% Pt-CuO improved the NO_2_ gas response but did not significantly improve the selectivity for other gases at the same time, so the selectivity of CuO to NO_2_ gas was effectively improved. As the response of 5 mol% Pt-CuO to HCHO is also greatly improved, the NO_2_ gas selectivity of Pt-CuO becomes worse.

Pristine CuO does not respond to NO_2_ gas at 25 °C. However, both Pt@CuO and Pt-CuO can respond well to NO_2_ gas at 25 °C due to the favorable catalytic and sensitization effects of Pt nanoparticles on NO_2_ gas. As shown in Figure 8, the responses of Pt@CuO and Pt-CuO conform to the power-law relationship with the NO_2_ gas concentration. Compared with Pt-CuO, Pt@CuO has a better response to NO_2_ gas at 25 °C. The responses of 2 mol% Pt @ CuO to 1, 2, 5, 10, and 20 ppm NO_2_ gas were 2.8, 3.5, 4.1, 5.2, and 5.8; the power-law relation formula is expressed as ${S=2.876\;C}^{0.24}$, where S is the response, and C is the NO_2_ gas concentration. The responses of 5 mol% Pt-CuO to 1, 2, 5, 10, and 20 ppm NO_2_ gas are 1.2, 1.7, 2, 2.2, and 2.4, respectively. The response of the gas-sensitive materials in relation to the concentration of NO_2_ gas conforms to the power-law equation, and the formula is expressed as ${S=1.359\;C}^{0.205}$.

Table 1 shows the reported sensing performance of various noble-metal-functionalized MOS-based sensors toward NO_2_. It can be clearly seen that both Pt-CuO and Pt@CuO in this work exhibit outstanding comprehensive performance with good responses and short response/recovery times for low concentrations of NO_2_. In addition, Figure 9 compares the reproducibility curves of 2 mol% Pt-CuO and 2 mol% Pt@CuO for (1–5 ppm) NO_2_ gases at 25 °C. All of the materials show consistent resistance change reproducibility, that is, good detection reversibility.

CuO functionalized with 2 mol% Pt exhibits excellent resistance to humidity interference when used for NO_2_ detection at 25 °C, as illustrated in Figure 10. The influence of humidity on the response is negligible. At relative humidity levels of 30%, 60%, and 90%, CuO loaded with Pt demonstrates responses of 3.01, 3.47, and 2.58 toward 2 ppm NO_2_, while responses of 1.33, 1.28, and 1.65 are obtained by Pt-doped CuO. Furthermore, the response and recovery times remain relatively stable across different humidity levels for both functionalization modes of CuO toward 2 ppm NO_2_, as shown in Figure 10b,c. The response and recovery curves for both functionalization modes of CuO toward 2 ppm NO_2_ at different humidities are shown in Appendix A.

### 3.3. Gas-Sensing Mechanism

The effect of noble metal functionalization can be explained by a combination of electronic and chemical sensitization [40]. For chemical sensitization, the active sites of noble metals on the surface of the metal oxide semiconductor are catalytically active for specific gases [41]. When gas molecules pass through, they first adsorb and catalyze on the surfaces of noble metals and then diffuse to the adjacent metal oxide semiconductor surface for the reaction, i.e., the spillover effect [42,43]. The diffusion of gas molecules on the material surface is facilitated by chemical sensitization, and the gas-sensitive reaction rate is accelerated at the same time. For electron sensitization, noble metals and metal oxides can accelerate the migration of carriers between them due to the different work functions, which increases the number of carriers involved in the gas-sensitive reaction, thus enhancing the response [44,45].

For Pt-loaded CuO, chemical sensitization plays a greater role, and for Pt-doped CuO, the opposite is true. Since most of the Pt deposited in Pt-loaded CuO forms particles on the CuO surface, the gas-sensitive response of the surface is closely related to its sensitivity. As the relative adsorbed oxygen content, based on XPS calculations, on the Pt-loaded CuO surface at room temperature is higher than that on Pt-doped CuO and the electron affinity for NO_2_ is stronger than that for oxygen, there will be more NO_2_ molecules adsorbed onto the Pt-loaded CuO surface at room temperature [46]. Thus, the Pt-loaded CuO sample exhibited good room-temperature gas-sensing performance.

On the other hand, the resistance of 2 mol% Pt@CuO decreases as the operating temperature increases, as shown in Appendix A, but in the case of 2 mol% Pt-CuO, the resistance at 100 °C is higher than that at 50 °C, as shown in Appendix A. This is mainly because the carrier mobility is limited by phonon scattering at 100 °C for the doping mode of CuO [47]. At lower temperatures, impurity scattering plays a dominant role, and the carrier mobility increases with the temperature [48]. The carrier concentration will be increased due to the thermal activation effect at higher temperatures, allowing an increase in conductivity [49].

However, changes in the band gap of noble-metal-functionalized semiconductors can alter the sensitivity of gas detection, as the concentration of electrons involved in gas-sensing reactions also plays an important role. At low temperatures, due to the fact that the carriers of p-type CuO are holes, the hole accumulation layer effect related to the band gap plays a dominant role [50]. The larger the band gap, the more pronounced the effect, resulting in a higher response of Pt-loaded CuO within 50 °C compared to Pt-doped CuO. As the operating temperature further increases, the narrow band gap formed by 2 mol% Pt-CuO leads to an increase in the electron concentration for reacting with NO_2_ gas, and the response of 2 mol% Pt-CuO to NO_2_ gas is significantly improved due to the increase in the carrier concentration and mobility [51,52]. The variation in band gap for different functionalization modes is demonstrated in Figure 11, which shows that the band gaps are 1.29 and 1.23 eV for 2 mol% Pt@CuO and 2 mol% Pt-CuO, respectively. The band gap reduction was caused by the formation of impurity bands in the band gap, and the Pt-doping mode can make the band gap narrower [53].

It should be noted that too high a temperature is not conducive to an improvement in the response, as the gas desorption rate also increases [54]. It can be inferred that the effect on the response is greater for lattice-doped Pt compared to surface-loaded Pt, but the required operating temperature of lattice-doped Pt is higher [55,56]. The FTIR patterns of pristine, Pt-loaded, and Pt-doped CuO nanorods are shown in Appendix A. Among them, the stretching vibration of the Cu–O bond induced three absorption peaks at 575, 490, and 440 cm^–1^ in the spectrum of the 2 mol% Pt-CuO sample, which all showed different degrees of redshift compared to those of the pristine CuO nanorods, and the degree of redshift was more intense than that of the 2 mol% Pt@CuO. This indicates that Pt doping is more likely to make the Cu-O bond weaker, thus corroborating the more intense change in the band gap.

## 4. Conclusions

CuO nanomaterials functionalized by Pt loading and doping were prepared by using a hydrothermal method with different timings of chloroplatinic acid addition. The functionalization of Pt significantly enhances the sensitivity and selectivity of CuO toward NO_2_ at low temperatures, with improvements in the response and recovery due to the chemical sensitization and electronic sensitization of the noble metal Pt. Pt-loaded CuO has more gas adsorption sites, leading to better chemical sensitization, which enhances the gas response of CuO to NO_2_ at low temperatures. On the other hand, Pt-doped CuO exhibits a narrower band gap and a closer interaction between Pt and CuO, producing more significant electronic sensitization that lowers the resistance of Pt-doped CuO at 25 °C, with more significant response changes as the temperature rises. This research provides a promising strategy for designing high-performance, low-temperature NO_2_ sensors for p-type CuO materials by selecting different functionalization modes.

## Figures and Tables

**Figure 1 sensors-24-00120-f001:**
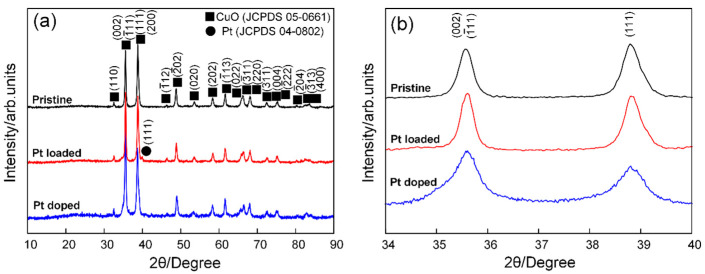
(**a**) XRD patterns of pristine CuO, 2 mol% Pt-CuO, and 2 mol% Pt@CuO; (**b**) magnified region of three main CuO diffraction peaks.

**Figure 2 sensors-24-00120-f002:**
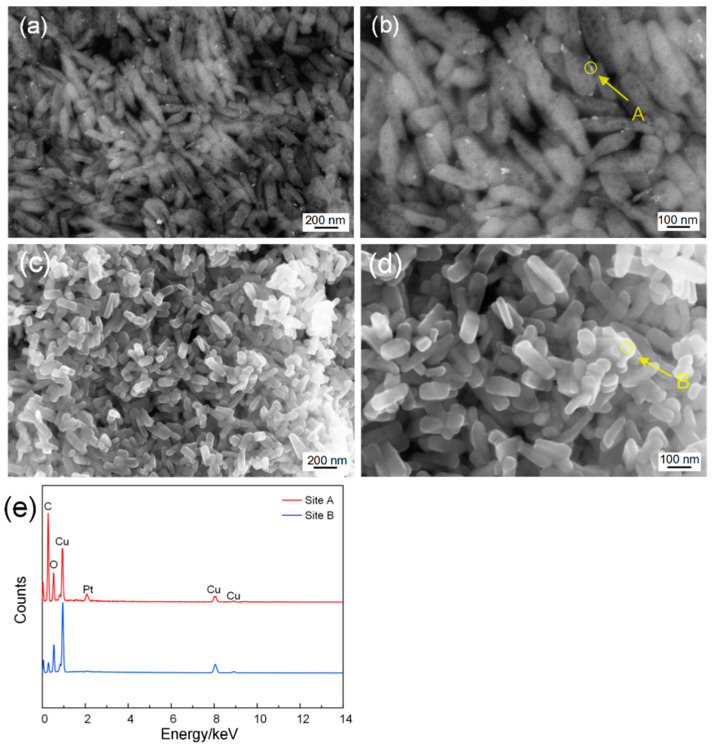
SEM images of 2 mol% Pt@CuO (**a**,**b**) and 2 mol% Pt-CuO (**c**,**d**); (**e**) the related EDS patterns.

**Figure 3 sensors-24-00120-f003:**
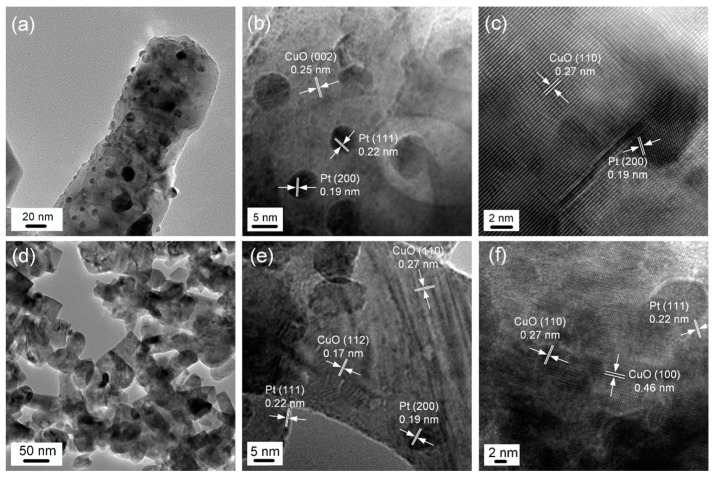
TEM images of 2 mol% Pt@CuO (**a**–**c**) and 2 mol% Pt-CuO (**d**–**f**).

**Figure 4 sensors-24-00120-f004:**
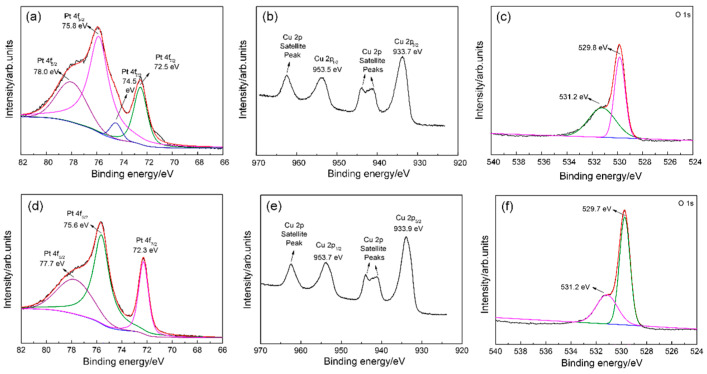
XPS spectra of Pt@CuO and Pt-CuO. (**a**,**d**) Pt 4f, (**b**,**e**) Cu 2p; (**c**,**f**) O 1s.

**Figure 5 sensors-24-00120-f005:**
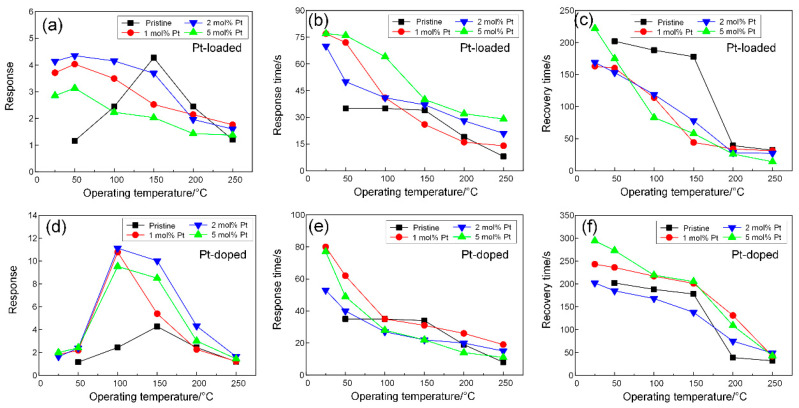
(**a**,**d**) Responses, (**b**,**e**) response times, and (**c**,**f**) recovery times of sensors based on Pt@CuO and Pt-CuO with different Pt concentrations when exposed to 5 ppm NO_2_ as a function of operating temperature.

**Figure 6 sensors-24-00120-f006:**
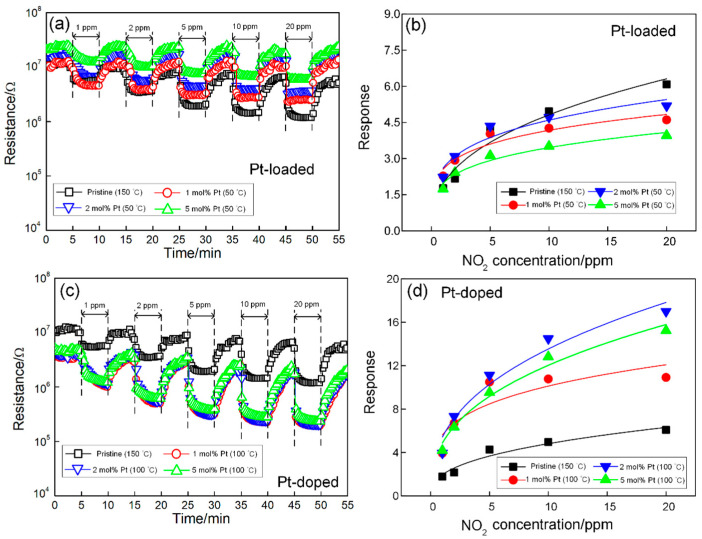
(**a**,**c**) Response–recovery curves of the sensors based on pristine CuO, Pt@CuO and Pt-CuO with different Pt concentrations to NO_2_ at respective optimal operating temperature. (**b**,**d**) The evolution of sensor response on the function of NO_2_ concentration.

**Figure 7 sensors-24-00120-f007:**
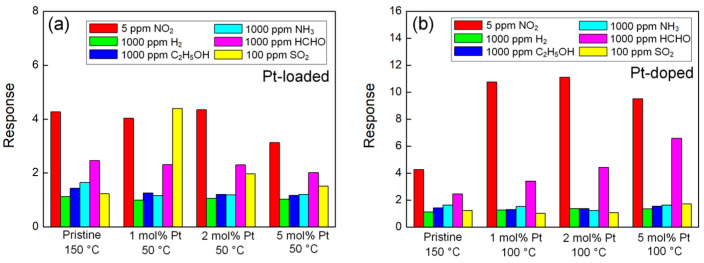
Responses of sensors based on Pt@CuO (**a**) and Pt-CuO (**b**) with different Pt concentrations to various gases at their respective optimal operating temperatures.

**Figure 8 sensors-24-00120-f008:**
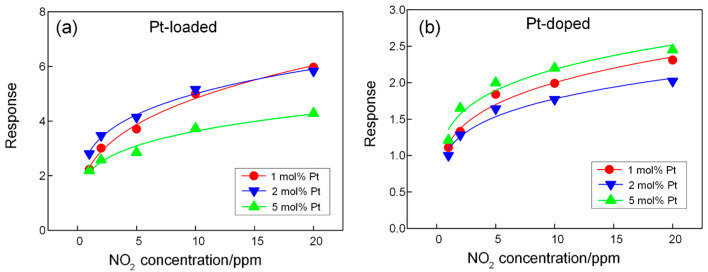
Responses of sensors based on Pt@CuO (**a**) and Pt-CuO (**b**) with different Pt concentrations to NO_2_ at 25 °C.

**Figure 9 sensors-24-00120-f009:**
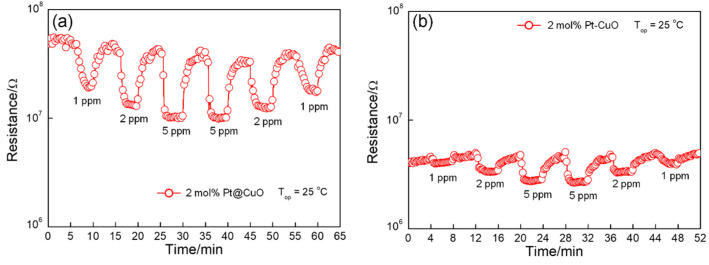
Reproducible cycles of gas sensors based on Pt@CuO (**a**) and Pt-CuO (**b**) upon exposure to NO_2_ at 25 °C.

**Figure 10 sensors-24-00120-f010:**
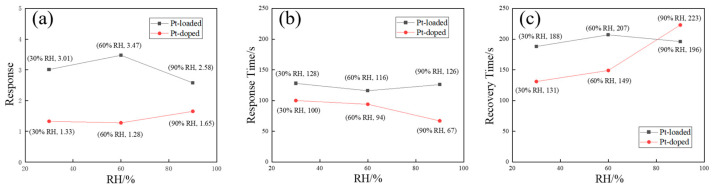
(**a**) Responses, (**b**) response times, and (**c**) recovery times of sensors based on Pt@CuO and Pt-CuO at different humidity levels when exposed to 2 ppm NO_2_ at 25 °C.

**Figure 11 sensors-24-00120-f011:**
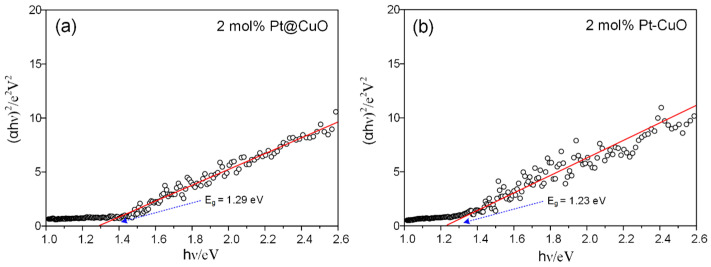
Plots of (αhν)^2^ versus the energy band gaps of Pt@CuO (**a**) and Pt-CuO (**b**).

**Table 1 sensors-24-00120-t001:** Comparison of the sensing performance of noble-metal-functionalized MOS-based sensors toward NO_2_.

Sensing Material	Operating Temperature/°C	Concentration/ppm	Response	Response Time/s	Recovery Time/s
Au@CuO nanorods [21]	100	5	4.3	-	-
Ag-CuO nanobricks [26]	22	20	67.2	35	900
Pd-WO_3_ nanoplates [38]	150	5	283.96	26	66
Au@WO_3_ core–shell nanospheres [39]	100	5	88	4	59
Pt-CuO nanorods in this work	25	2	1.33	125	188
Pt@CuO nanorods in this work	25	2	3.01	100	131

## Data Availability

Data will be made available on request.

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
