# Peer review of "Influence of Different Pt Functionalization Modes on the Properties of CuO Gas-Sensing Materials"

_sensors, 2023, doi:10.3390/s24010120_

Round 1

Reviewer 1 Report

Comments and Suggestions for Authors

In this research, the authors explore the impact of different functionalization modes, specifically noble metal loading and doping, on the sensitivity and selectivity of metal oxide semiconductor (MOS) based gas sensors, with a focus on CuO nanomaterials. They synthesized Pt-doped CuO and Pt-loaded CuO using a one-pot hydrothermal method and dipping method, respectively. The study reveals that both functionalization methods significantly enhance the sensitivity and selectivity of CuO to NO2 at low temperatures. Interestingly, CuO with Pt-loading exhibits superior sensing performance at 25 degree C, while CuO with Pt-doping shows more substantial sensitivity changes with increasing temperature. The observed differences are attributed to distinct sensitization mechanisms: Pt-doping emphasizes electron sensitization, while Pt-loading emphasizes chemical sensitization. Pt-loaded CuO, with more gas adsorption sites, benefits from enhanced chemical sensitization, leading to improved gas sensitivity at low temperatures. Conversely, Pt-doped CuO exhibits a narrower bandgap and stronger electronic sensitization, lowering resistance at 25 °C with significant sensitivity changes at higher temperatures. The findings contribute innovative insights into optimizing noble metal functionalization for gas sensing in metal oxide semiconductors. The research proposes a promising strategy for designing high-performance low-temperature sensors for NO2 using different functionalization modes on p-type CuO materials.

Comment:

The selectivity limitations of αhν stem from its exclusive emphasis on photon absorption, neglecting interactions with other electronic states in the material. It fails to account for processes like phonon scattering and impurity effects, which can impact electronic transitions. Additionally, αhν lacks information about the relative positions and interactions between electronic states in different energy bands. In summary, while αhν provides insights into absorption, its selectivity assessment is constrained by its narrow focus on absorption events. The author can expand the discussion.

Author Response

Dear Reviewer,

The reviewers’ comments to our manuscript (sensors-2770854) submitted to “SENSORS” have been received.We would like to express our sincere appreciation for your considerate work on this article, and acknowledge the comments provided by you.We have addressed all the comments. Please see the attachment.

Please do not hesitate to contact us if you require more information. Thank you very much again for your help.

Sincerely yours,

Yanbai Shen

School of Resources and Civil Engineering, Northeastern University, Shenyang 110819, China

Reviewer 2 Report

Comments and Suggestions for Authors

General comments:

The work compares the performance of two functionalization modes of CuO based gas sensor with Pt, aiming at high sensitivity sensors for No2 detection.

The manuscript is well structured and describes the synthesis and characterization of sensing materials, as well as the characterization of the sensors into NO2 environment taking into account several important aspects.

Authors should include some comparison of their results with the literature to help the reader verify the impact of the proposal.

Specific comments:

- Lines 261-262: That first phrase in item 3.2 is strange and can be removed.

- Lines 287-288: There are some repetitions about “2mol% Pt doping”. Please rewrite that phrase.

- Line 346: Please correct: replace “Au” with “Pt”.

Comments on the Quality of English Language

A careful review of English would be good.

Author Response

(The authors gave the same response as above.)
